# Moderating Effects of Structural Empowerment and Resilience in the Relationship between Nurses’ Workplace Bullying and Work Outcomes: A Cross-Sectional Correlational Study

**DOI:** 10.3390/ijerph18041431

**Published:** 2021-02-03

**Authors:** Heiyoung Kang, Kihye Han

**Affiliations:** 1Department of Nursing, Seoul National University Hospital, Seoul 03080, Korea; queen-big@hanmail.net; 2College of Nursing, Chung-Ang University, Seoul 06974, Korea

**Keywords:** workplace bullying, work outcomes, structural empowerment, resilience

## Abstract

This cross-sectional correlational study aimed to evaluate the moderating effects of structural empowerment and resilience in the relationship between workplace bullying and nursing work outcomes. Data were collected from 435 nurses and nurse managers working at a tertiary hospital in Seoul, South Korea. The moderating effects were examined using stepwise hierarchical multiple regression models. The bootstrapping method was utilized to verify the magnitude and significance of the moderating effects. Structural empowerment showed a moderating effect in the relationship between workplace bullying and nursing work outcomes: for the conditional values above the average level of structural empowerment, workplace bullying was significantly associated with nursing work outcomes, while work outcomes were low regardless of workplace bullying for the conditional values less than average. However, resilience had no moderating effect. To improve work outcomes, bullying must be reduced and structural empowerment and resilience increased.

## 1. Introduction

### 1.1. Workplace Bullying

Uncivil workplace behaviors often arise in organizations with job-related tensions and conflicts, and include insults, pressure, and refusal to talk, known as mobbing [1]. Workplace bullying stems from power differences within an organization. Unlike horizontal harassment, which appears in relation to gender, religion, age, race, and disability [2], workplace bullying is a vertical type, occurs more frequently, and lasts longer [3]. Group members often remain silent about perpetrators’ behaviors if they show high work performance and are powerful [4]. They treat bullying of the underprivileged as a means of socializing and training new members into their organization. In a rigid organizational culture, the prevalence of reported bullying tends to be much lower than the actual occurrence [5]. According to the Korea Vocational Competency Development Institute, the nursing profession ranked highest (41.3%) in the prevalence of workplace bullying [6].

### 1.2. Nursing and Workplace Bullying

In the authoritative hierarchical structures of hospitals, nurses are exposed to various stressors, such as pressure regarding productivity and cost reduction, contact with patients, and coordination with other departments and professions [7]. As they often cannot appropriately digest stress and internally suppress their feelings of low self-esteem, inferiority, and hatred, they often express these feelings through the oppression and bullying of colleagues with less power than themselves [8]. Workplace bullying mentally damages the affected nurses and impairs work performance, which has negative effects on patient care [9]. Unfavorable nursing work environments with high workloads and bullying can negatively affect nursing work outcomes, such as medication errors or falls [10].

### 1.3. Workplace Bullying and the Job Demands–Resources Model

The phenomena of workplace bullying can be explained by the job demands–resources model [11]. This model divides organization’s job characteristics into job demands and job resources. Job resources can lead to outstanding work performance by inducing organizational commitment through a motivational process. Job demands deplete energy and cause health problems through a health impairment process [12,13,14], and consequently have a large impact on the organization’s outcomes. When job resources (e.g., supervisor support, work autonomy, performance feedback, etc.) are lacking and job demands (e.g., excessive workload) are high, workplace bullying is more likely to occur [15]. Poorly designed work conditions and chronic job demands such as bullying can have a significant impact on work outcomes.

### 1.4. Structural Empowerment and Resilience

At the organizational level, structural empowerment is an important job resource. It refers to the level of accessibility to learning and growth opportunities, information, advocacy, and resources for effective job performance [16]. When there is a high level of structural empowerment, nurses can easily access organizational resources to control stressful situations (e.g., bullying) and would not experience significant negative health or work performance consequences [17]. At a low level of structural empowerment, the bullied individual does not have organizational support to overcome stressors such as workplace bullying, which can threaten their physical and mental health and adversely affect patient care.

Resilience is an individual-level competency and a positive factor in overcoming adversity, conflict, failure, and increased responsibility [18]. Resilience is an essential attribute because it promotes the use of individual internal and external resources to overcome difficult situations, reduces the negative consequences of workplace bullying, and supports good work outcomes [19,20]. Nurses with strong resilience have the assertiveness to overcome and resist workplace bullying [21]. On the other hand, less resilient nurses cannot tolerate bullying and may therefore perform poorly.

### 1.5. Study Aims

Few studies examine the moderators of workplace bullying. The job demands–resources model classifies job characteristics into two dimensions (job demands and job resources), but does not consider individual job aspects. This study investigated the moderating effects of structural empowerment at the organizational level and resilience at the individual level in the relationship between workplace bullying and nursing outcomes (Figure 1). Our study hypotheses were that workplace bullying experience would be negatively associated with work outcomes, and that the relationship between workplace bullying and work outcomes would be different according to the levels of structural empowerment and resilience.

## 2. Materials and Methods

### 2.1. Design

This study employed a cross-sectional survey design.

### 2.2. Participants and Procedures

The subjects of this study were nurses and nursing managers with more than six months of clinical experience working in a tertiary hospital in Seoul. For the data collection, 41 units were conveniently selected based on the types of units (34 general wards and 7 specialty units). The questionnaire packages were distributed to each unit. Nurses who voluntarily agreed to participate in the study completed the questionnaire anonymously at any place they wanted. It took 15–20 min to complete the questionnaire. The nurse participants returned them in a sealed envelope to designated return boxes. A total of 520 survey questionnaires were distributed, 455 of which were returned (return rate = 87.5%). We excluded 20 cases with missing data in the study key variables (i.e., experience of workplace bullying, work outcomes, structural empowerment, resilience), and used 435 cases for analysis. Before data collection, the questionnaire and the study procedure were reviewed and approved by the Medical Research Ethics Review Committee of the study hospital (IRB approval number: H-1808-161-967).

### 2.3. Measures

#### 2.3.1. Experience of Workplace Bullying

Workplace Bullying in Nursing-Type Inventory (WPBN-TI) was used to measure the experience of workplace bullying [22,23]. WPBN-TI has 16 Likert-type items (ranging from ‘not at all’ (1) to ‘very much’ (4)), with three subdomains of verbal and non-verbal bullying, work-related bullying, and external threats. The overall scale score was calculated as the average of the 16 scores, with higher scores indicative of experiencing workplace bullying more frequently (Cronbach’s α = 0.92).

#### 2.3.2. Structural Empowerment

To measure structural empowerment, the Condition of Work Effectiveness Questionnaire was utilized [24]. The Korean version of the Condition of Work Effectiveness Questionnaire has demonstrated reliable (Cronbach’s alphas ranged from 0.69 to 0.89 across the sub-domains) and had construct validity (Lee, E.S., 2016). This survey consists of 12 items, which are measured on a Likert scale ranging from ‘not at all’ (1) to ‘very much’ (5). The overall scale score was calculated as the average of the item scores. Higher scores indicate a higher level of structural empowerment (Cronbach’s α = 0.86 in our study).

#### 2.3.3. Resilience

The Korean version of the Connor-Davidson Resilience Scale (K-CD-RISC), whose reliability and validity have been verified, was used to measure resilience [25]. Each item was measured on a Likert scale ranging from ‘not at all’ (1) to ‘very much’ (5), and the overall scale score was calculated as the average of the item scores. Higher scores mean a higher resilience (Cronbach’s α = 0.92).

#### 2.3.4. Work Outcomes

Nursing work outcomes were measured by the Nursing Performance Appraisal Scale [26,27,28]. It is composed of 24 questions in five subdomains of nursing performance ability (seven items), attitude to nursing (six items), improving the level of nursing care (five items), nursing process application (three items), and nursing support function (three items). Each item was measured on a Likert scale ranging from ‘not at all’ (1) to ‘very much’ (5), and the overall scale score was calculated as the average of the item scores. The higher the score, the better the nursing work outcome (Cronbach’s α = 0.94).

#### 2.3.5. General Characteristics

We collected data on participant characteristics including age, education, marital status, position, years of registered nurse (RN) experience, years at the current workplace, and work department. We also inquired about experiences witnessing bullying during the total working period.

Subjective workload and social support were the controlling variables we measured, as they are reportedly significantly associated with nurses’ work performance and outcomes [29]. The subjective workload was assessed for quantitative demands (four items), work pace (three items), and emotional demands (four items) at work, which were measured using subdomains of the Korean version of the Copenhagen Psycho-social Questionnaire II [29]. Each item was measured on a Likert scale ranging from ‘not at all’ (1) to ‘always’ (5), and the overall scale score was calculated as the average of the item scores. Higher scores indicate a more subjective workload (Cronbach’s α = 0.79). We used Park’s social support survey scale [30] to measure social support. This survey consists of 23 items, each measured on a Likert scale ranging from ‘not at all’ (1) to ‘always’ (4). The overall scale score was calculated as the average of the item scores. The higher the score, the better the social support (Cronbach’s α = 0.97).

### 2.4. Statistical Analysis

The general participant characteristics were analyzed using frequencies and percentages. Experiences with bullying, nursing work outcomes, structural empowerment, and resilience were analyzed using means and standard deviations. *T*-tests and ANOVAs were performed with the Scheffe post hoc test to examine differences in bullying experience, structural empowerment, resilience, and nursing work performance by general characteristics. We generated multiple regression models to investigate the effect of experiencing bullying on nursing work outcomes. The moderating effects of structural empowerment and resilience on the relationship between workplace bullying and work outcomes were initially examined in a stepwise hierarchical manner. The magnitude and significance of the moderators (i.e., structural empowerment and resilience) were verified using the SPSS macro and bootstrapping method proposed by Hayes [31]. All statistical analysis was conducted using SPSS/WIN 25.0 (IBM Corp., Armonk, NY, USA).

## 3. Results

### 3.1. Participant Characteristics

The average participant age was 32.64 ± 6.85 years (Table 1). Most of the participants were female (94%), single (72%), and staff nurses (90%). During the entire working period, 69% of the nurses had witnessed bullying behaviors perpetrated by others to others at their workplaces.

The mean scores were 3.34 (standard deviation [SD] = 0.55) for the level of subjective workload, 2.82 (SD = 0.45) for social support, 1.84 (SD = 0.58) for workplace bullying experience, 3.77 (SD = 0.49) for nursing work outcomes, 3.09 (SD = 0.54) for structural empowerment, and 3.34 (SD = 0.48) for resilience. Work-related bullying was the most frequently reported type of bullying (mean ± SD = 2.09 ± 0.71), followed by verbal/nonverbal bullying (mean ± SD = 1.84 ± 0.69) and external threats (mean ± SD = 1.27 ± 0.51). Regarding nursing work outcomes, nursing performance ability was highest (mean ± SD = 3.99 ± 0.51), followed by nursing support function (mean ± SD = 3.91 ± 0.61), attitude towards nursing (mean ± SD = 3.88 ± 0.56), the application of the nursing process (mean ± SD = 3.71 ± 0.71), and improving the level of nursing care (mean ± SD = 3.28 ± 0.81).

### 3.2. Experiences of Workplace Bullying

Differences in workplace bullying experience were statistically significant according to age, marital status, years of service, work department, and having witnessed bullying during the total working period (Table 2). Nurses under the age of 29 reported significantly more frequent workplace bullying experiences than those aged 30–39 years (F = 4.703, *p* < 0.01). Those with less than one to three years of RN experience reported more frequent workplace bullying than those with five to 10 years of experience (F = 4.488, *p* < 0.01). Nurses working in the operating room had more bullying experience than those working in general wards or emergency rooms (F = 6.93, *p* < 0.01).

### 3.3. The Moderating Effects of Structural Empowerment and Resilience on the Relationship between Bullying and Nursing Work Outcomes

In the initial stepwise regression models, workplace bullying did not have a significant effect on nursing work outcomes (β = −0.033, *p* > 0.05) (Table 3, Step 1). Higher structural empowerment (β = 0.300, *p* < 0.001) (Table 3, Model 1, Step 2) and higher resilience (β = 0.378, *p* < 0.001) (Table 3, Model 2, Step 2) had a significant effect on nursing work outcomes. The interaction variable between bullying and structural empowerment was not statistically significant (β = −0.068, *p* > 0.05) (Table 3, Model 1, Step 3), nor was the interaction variable between bullying and resilience (β = 0.010, *p* > 0.05) (Table 3, Model 2, Step 3).

The results from the bootstrapping method analysis are presented in Table 4.

Where the structural empowerment was average or above (i.e., average + 1SD), those with higher levels of workplace bullying experience were significantly more likely to report lower nursing work outcomes (B = −0.138, 95% CI = −0.215~0.061 for average structural empowerment; B = −0.203, 95% CI = −0.308~0.097 for high structural empowerment). On the other hand, where the structural empowerment was below average (i.e., average − 1SD), the nursing outcome was low, regardless of workplace bullying experience (B = −0.074, 95% CI = −0.175~0.028). Figure 2 shows the differences in the slope of the workplace bullying experience for work outcomes according to the conditional value of structural empowerment. These differences suggested a moderating effect of structural empowerment.

There was no significant difference in the effects of workplace bullying on nursing work outcomes by level of resilience.

## 4. Discussion 

Although bullying did not have a significant effect on nursing work outcomes, the two moderating variables of structural empowerment and resilience had direct effects on nursing work outcomes. Structural empowerment had a moderating effect on the relationship between bullying and nursing work outcomes. Our study suggests that organizational efforts are needed to maintain a high level of structural empowerment and to minimize workplace bullying, which can negatively affect nursing work outcomes.

### 4.1. Experiences of Workplace Bullying and Nursing Work Outcomes

The experience of workplace bullying in our study (mean score = 1.84) was slightly lower than that of a recent study conducted in a tertiary hospital in Korea (mean score = 2.00) [32]. Our study data were conveniently collected, which may have excluded some wards with poor circumstances or nurses who had experienced severe workplace bullying during the data collection period. Future studies should include a variety of hospitals, and follow-up research should have data collection based on a random sampling method.

Workplace bullying and the perception of a higher subjective workload and less social support was more often experienced by nurses working in operating rooms and intensive care units than those in wards and emergency rooms. Under unfavorable work conditions with a high workload, nurses cannot adequately control their stress and emotion and may demonstrate aggressive and offensive behavior [8]. Nurses aged 50 years or older reported having frequently experienced workplace bullying. These findings suggest that workplace bullying may be caused by a power imbalance [33] and may occur regardless of age and position. This is in contrast with the perception that the victim is helpless and the perpetrator is of higher status than the victim. Although racism and gender/sexual orientation-based discrimination are known to influence workplace bullying [34], these factors are not as important in the homogeneous Korean nursing community.

Work-related bullying was the most frequently reported type of bullying, whereas the RN network survey reported verbal bullying as the most frequently experienced bullying [35]. This reflects the cultural differences in nursing work and organizational environments. As nursing work is directly connected to patient safety, thorough procedures are required to reduce errors [36]. Unstable nursing work environments with an increase in nursing workload or workplace bullying can negatively affect nursing work outcomes, such as medication errors or falls [10]. As nurses leave hospitals and nursing shortages grow, the amount of work that each nurse has to do grows, causing work-related bullying among nurses to increase [37]. Efforts to maintain proper nurse staffing and efficient task allocation must be made.

Contrary to the study hypothesis, the level of workplace bullying experience did not affect nursing work outcomes. This was consistent with previous research [15] in which bullying had a significant effect on job satisfaction, but not work outcomes. Workplace bullying did not directly affect the work outcomes because these outcomes were measured using self-appraisal items for performance ability, performance attitude, and level of nursing care. Although bullying experience did not directly affect work outcomes, workplace bullying leads to anxious and uncomfortable working environments, which can adversely affect patient care and nursing work outcomes [15]. According to a prospective study, long-term workplace bullying impairs the health of the victims and negatively affects work outcomes [38]. This suggests that organizational efforts to eradicate workplace bullying are necessary to increase nurses’ work outcomes.

### 4.2. The Moderating Effect of Structural Empowerment on the Relationship between Workplace Bullying and Nursing Work Outcomes

The associations between workplace bullying and work outcomes differed by level of structural empowerment. In the group with higher-than-average structural empowerment, negative slopes, or the negative relationship between bullying and nursing work outcomes, were evident. However, in the group with low structural empowerment, nursing work outcomes were low, regardless of the level of bullying experience, supporting the direct effect between structural empowerment and nursing work outcomes. We found that structural empowerment had a significant direct association with nursing work outcomes, per previous research [39,40,41]. Nurses who are empowered at work perceive inter-professional collaboration and work more effectively [17]. However, nurses with low levels of structural empowerment may not deliver quality care due to their inability to collaborate [17].

Organizations should empower their nurses by increasing learning and growth opportunities and access to information and resources. For sufficient and appropriate structural empowerment in nursing, authentic leadership is important [21,42]. Authentic leaders reinforce openness with their staff and provide them with opportunities to express ideas and opinions. This allows for a sense of community and job commitment throughout the organization [43]. Managers’ authentic leadership is important for young nurses who perceive a low level of structural empowerment, as they are more influential for nurses with a shorter tenure [44].

### 4.3. The Moderating Effect of Resilience on the Relationship between Bullying and Nursing Work Outcomes

Resilience did not moderate the relationship between bullying and work outcomes. Individuals with low resilience could be more vulnerable to workplace adversity [45,46]. Bullying behaviors bring about systemic problems that conflict with safe patient care and affect organizational outcomes at all levels of health care [47]. However, resilience had a positive association with nursing work performance, and increasing resilience helps nurses reduce their emotional burnout and increases job commitment. This helps to maintain high nursing work outcomes [48].

### 4.4. Research Limitations

The study findings should be interpreted with caution. As this study analyzed data on 435 nurses working at one tertiary university hospital, there are limitations in generalizing the research results to clinical nurses in all hospitals in Korea. Furthermore, the cross-sectional descriptive research design makes it impossible to derive a causal relationship between the experience of bullying and nursing work outcomes. All survey data were based on self-reported responses, and denial and social desirability might affect the study findings.

## 5. Conclusions

According to the hierarchy of controls framework, workplace bullying could be controlled step by step [49]. In the first step, workplace bullying could be reduced if organizations empower their workers and make them aware of the negative consequences of workplace bullying. Hospital organizations should provide nurse managers with education programs on the psychodynamics of bullying to promote a greater understanding of bullying in their units. For the second step, institutions should develop a periodic bullying monitoring system in which they guarantee anonymity and offer help when victims report workplace bullying. Sufficient staff support and proper human resource management should be provided in units with high patient acuity and nursing workloads where workplace bullying occurs frequently. To adequately compensate for the assigned workload, an appropriate wage system should be established as a tangible reward. Managers should be trained in establishing and applying workplace bullying management protocols [50]. In the US, the Workplace Bullying Institute (WBI) is dedicated to the eradication of workplace bullying. The WBI’s activities, including public education, training for professionals, unions, and employers, legislative advocacy, and consulting solutions for organizations, could be adapted for the Korean healthcare organizations’ culture. To promote structural empowerment, managers, seniors, and colleagues should share information, feedback, and support. Once the work environmental factors (e.g., work organization, workload, or adequate staffing) are addressed, individual-focused prevention measures could be considered. To increase nurses’ resilience, professional meetings would be useful to cultivate internal motivation to achieve organizational goals. Hospital-wide authentic leadership development programs could promote openness among nurses, and educational programs to increase nurses’ awareness of workplace bullying and to encourage them to discuss harmful situations may support resilience [51].

## Figures and Tables

**Figure 1 ijerph-18-01431-f001:**
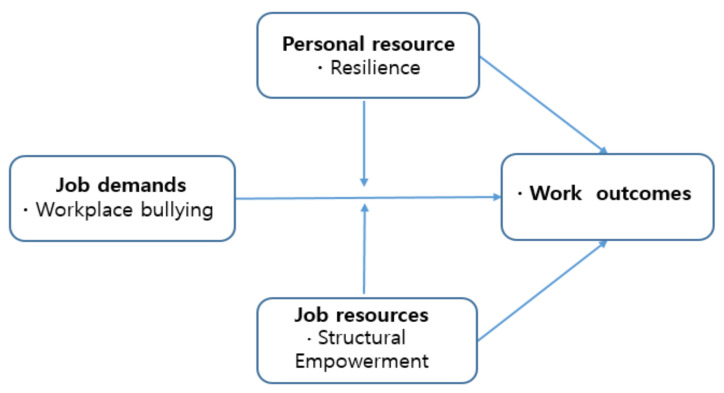
A conceptual framework based on the job demands–job resources model.

**Figure 2 ijerph-18-01431-f002:**
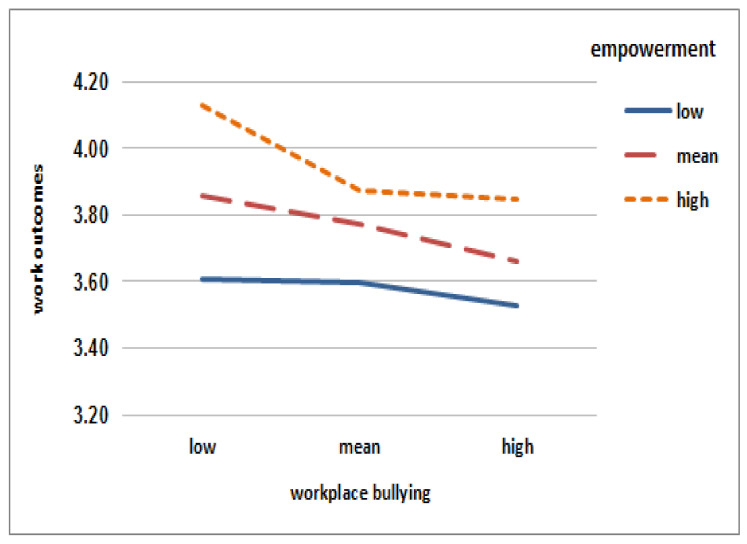
Simple slopes equations for the association between workplace bullying and nursing work outcomes by structural empowerment.

**Table 1 ijerph-18-01431-t001:** Participant characteristics (*n* = 435).

Variables	Classification	*n*	(%)	M ± SD	Range
Age	<29	190	(43.7)	32.64 ± 6.85	
	30–39	178	(40.9)		
	40–49	53	(12.2)		
	≥50	14	(3.2)		
Gender	Female	410	(94.3)		
	Male	25	(5.7)		
Marital status	Single	311	(71.5)		
	Married	124	(28.5)		
Position	Staff nurse	391	(89.9)		
	Charge nurse	29	(6.7)		
	Unit manager	15	(3.4)		
Years of work experience	1	25	(5.7)		
	1–<3	92	(21.1)		
	3–<5	97	(22.3)	7.49 ± 6.76	
	5–<10	104	(23.9)		
	≥10	117	(26.9)		
Years at current workplace	1	50	(11.5)		
	1–<3	132	(30.3)		
	3–<5	116	(26.7)	4.37 ± 3.89	
	5–<10	96	(22.1)		
	≥10	41	(9.4)		
Work department	Ward	225	(51.7)		
	Intensive care unit	122	(28.0)		
	Emergency room	34	(7.8)		
	Operating room	54	(12.4)		
Witnessed bullying during the total working period	Yes	300	(69.0)		
	No	135	(31.0)		
Subjective workload	Overall			3.34 ± 0.55	1–5
	Ward			3.28 ± 0.54	
	Intensive care unit			3.40 ± 0.52	
	Emergency room			3.28 ± 0.67	
	Operating room			3.48 ± 0.58	
Social support	Overall			2.82 ± 0.45	1–4
	Ward			2.87 ± 0.43	
	Intensive care unit			2.74 ± 0.45	
	Emergency room			2.97 ± 0.56	
	Operating room			2.69 ± 0.41	
Experience of workplace bullying	Overall			1.84 ± 0.58	1–4
	Work related bullying			2.09 ± 0.71	
	Verbal/non-verbal bullying			1.84 ± 0.69	
	External threats			1.27 ± 0.51	
	Overall			3.77 ± 0.49	1–5
Nursing work outcomes	Nursing performance ability			3.99 ± 0.51	
	Attitude towards nursing			3.88 ± 0.56	
	Nursing support function			3.91 ± 0.61	
	Improving the level of nursing			3.28 ± 0.81	
	Application of the nursing process			3.71 ± 0.71	
Structural empowerment				3.09 ± 0.54	1–5
Resilience				3.34 ± 0.48	1–5

**Table 2 ijerph-18-01431-t002:** Differences in experiences of workplace bullying by participant characteristics (*n* = 435).

Characteristics	Categories	Experience of Workplace Bullying		
		M ± SD	t/F	Scheffe
			(p)	
Age	<29 (a)	1.95 ± 0.59	4.703	b < a
	30–39 (b)	1.73 ± 0.54	(<0.01)	
	40–49 (c)	1.81 ± 0.59		
	≥50 (d)	1.87 ± 0.62		
Gender	Female	1.84 ± 0.58	−0.468	-
	Male	1.89 ± 0.59	(0.64)	
Marital status	Single	1.88 ± 0.57	2.119	-
	Married	1.75 ± 0.59	(0.04)	
Position	Staff nurse	1.84 ± 0.58	0.061	-
	Charge nurse	1.83 ± 0.59	(0.94)	
	Unit manager	1.79 ± 0.53		
Years of total RN experience	1 (a)	1.93 ± 0.62	4.488	d < b
	1–<3 (b)	2.04 ± 0.55	(<0.01)	
	3–<5 (c)	1.84 ± 0.61		
	5–<10 (d)	1.71 ± 0.56		
	≥10 (e)	1.79 ± 0.56		
Years at current workplace	1(a)	1.82 ± 0.58	2.127	-
	1–<3 (b)	1.93 ± 0.58	(0.08)	
	3–<5 (c)	1.82 ± 0.59		
	5–<10 (d)	1.72 ± 0.55		
	≥10 (e)	1.91 ± 0.58		
Work department	Ward (a)	1.75 ± 0.55	6.93	a, c < d
	Intensive care unit (b)	1.91 ± 0.58	(<0.01)	
	Emergency room (c)	1.74 ± 0.64		
	Operating room (d)	2.11 ± 0.58		
Witnessed bullying during the total working period	Yes	1.95 ± 0.58	6.043	-
	No	1.60 ± 0.50	(<0.01)	

**Table 3 ijerph-18-01431-t003:** Associations between the experience of workplace bullying (X), structural empowerment (M1)/resilience (M2), and nursing work outcomes (Y).

Explanatory Variables	Nursing Work Outcomes					
	Step 1		Step 2		Step 3	
	β	t	β	t	β	t
Model 1						
Experience of workplace bullying (X)	−0.033	−0.626	−0.009	−0.175	−0.012	−0.235
Structural empowerment (M1)			0.300	6.108 ***	0.306	6.231 ***
X * M1					−0.068	−1.649
Model 2						
Experience of workplace bullying (X)	−0.033	−0.626	−0.057	−1.167	−0.058	−1.18
Resilience (M2)			0.378	8.884 *****	0.375	8.646 ***
X * M2					0.010	0.239

***, *p* < 0.001; Models were adjusted for age, marital status, total years of service, current work experience, subjective workload, and social support.

**Table 4 ijerph-18-01431-t004:** The moderating effect of structural empowerment and resilience according to conditional values in the relationship between workplace bullying and nursing work outcomes (*n* = 435).

	Nursing Work Outcomes		
Moderator	B	Boot SE	95% CI of B
Structural empowerment			
Low (mean − 1SD)	−0.074	0.051	−0.175~0.028
Mean	−0.138	0.039	−0.215~−0.061
High (mean + 1SD)	−0.203	0.054	−0.308~−0.097
Resilience			
Low (mean − 1SD)	−0.190	.049	−0.287~−0.093
Mean	−0.173	.036	−0.244~−0.102
High (mean + 1SD)	−0.156	.046	−0.246~−0.065

SD = standard deviation.

## Data Availability

Due to the nature of this research, participants of this study did not agree for their data to be shared publicly, so supporting data is not available.

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
