# Peer review of "Moderating Effects of Structural Empowerment and Resilience in the Relationship between Nurses’ Workplace Bullying and Work Outcomes: A Cross-Sectional Correlational Study"

_ijerph, 2021, doi:10.3390/ijerph18041431_

Round 1

Reviewer 1 Report

I have read with interest the paper entitled ‘Nurses’ workplace bullying and work outcomes: moderating effects of structural empowerment and resilience’.

The paper is well written and quite easy to read. Below are my comments on the manuscript.

- There is a distinction between “vertical” bullying (from someone in a powerful position towards a collaborator) and “horizontal” bullying between workers at a similar hierarchical rank.

- Authors need to discuss (even shortly) how racism, and gender/sexual orientation-based discrimination may influence workplace bullying. Do authors think that workplace bullying encompasses these notions? The explanation through the job-demands model seems to explain just a portion of the overall phenomenon.

- I would be careful on the use of the word “victim” (page 2, row 59), which has more to do with legal matters than psychological/medical/scientific matters.

- How does the concept of resilience fit (or does not fit) with the hierarchy of controls framework (or equivalent regulations, such as general principles of prevention in the EU) for the prevention of occupational risks at the workplace? https://www.cdc.gov/niosh/topics/hierarchy/default.html
Resilience is an equivalent of an individual-focused prevention measure that should be considered after addressing the work environment factors (work organization, workload, adequate staffing etc.)

- I am surprised that authors state that few studies have investigated the moderators of workplace bullying. Were synonyms or similar concepts (mobbing, harassment) considered in the research?

- Can authors clarify (row 152) “the nurses had witnessed bullying”: does it include experiencing bullying and witnessing bullying behaviors perpetrated by others to others?

- Presentation of data on rows 153 is a bit difficult to read. Is it possible to have the number and the outcome next one to each other?

- Table 1 would be easier to read with horizontal lines as separators between each variable.

- How was the Condition of Work Effectiveness Questionnaire used in the Korean context. Was there a Korean version available? The questionnaire has 12 items, each comprising many sub-items.

- Authors state (row 275) that individuals with low resilience are more easily involved and victimized in bullying. Is that data based on longitudinal studies? We can also think that bullied people have a decrease in their resilience level.

- Where were the questionnaires filled? The location may influence results (at the workplace vs. at home).

- Do we have the evidence that providing training on bullying management protocols is effective? Or is that a speculation based on the authors’ work?

- I would like some more description on how the work conditions are assessed, going above the nature of relationships between workers (supplies, medication, beds, patient flows etc.) All of this has a strong influence on psychosocial risks. The Structural Empowerment questionnaire assessed this dimension a little bit, but this seems very limited to me. My understanding is that authors put the emphasis on resilience, training on bullying, bullying monitoring system, education programs (although they mention the need for sufficient staff support).

- All in all, the questionnaires may be quite long to fill. How long was that? Can authors expand a bit more on missing data (row 86): what is the threshold for them to exclude the 20 cases (e.g. a single non answered question)?

Author Response

Response to Reviewer 1 Comments

Comments and Suggestions for Authors

I have read with interest the paper entitled ‘Nurses’ workplace bullying and work outcomes: moderating effects of structural empowerment and resilience’. The paper is well written and quite easy to read. Below are my comments on the manuscript.

  1. There is a distinction between “vertical” bullying (from someone in a powerful position towards a collaborator) and “horizontal” bullying between workers at a similar hierarchical rank.

Thank you for your comment. We expanded the introduction section to include vertical vs horizontal bullying (lines 25-30).

  1. Authors need to discuss (even shortly) how racism, and gender/sexual orientation-based discrimination may influence workplace bullying. Do authors think that workplace bullying encompasses these notions? The explanation through the job-demands model seems to explain just a portion of the overall phenomenon.

Thank you for your comment. We expanded the introduction section to include vertical vs horizontal bullying (lines 25-30), and added the discussion about racism and gender/sexual orientation in the workplace bullying (lines 248-250).

  1. I would be careful on the use of the word “victim” (page 2, row 59), which has more to do with legal matters than psychological/medical/scientific matters.

Thank you for pointing this out. We changed the term “victim” to “the bullied” (lines 64-65).

  1. How does the concept of resilience fit (or does not fit) with the hierarchy of controls framework (or equivalent regulations, such as general principles of prevention in the EU) for the prevention of occupational risks at the workplace? https://www.cdc.gov/niosh/topics/hierarchy/default.html

Resilience, a type of individual-level resources, would change attitudes toward the adversity and help overcome the conflict. The hierarchy of controls framework focuses on the organizational-level physical controlling of exposure to occupational hazard and is known as a useful means of determining how to implement feasible and effective control solutions. Although resilience does not fit with the hierarchy of controls framework, we included this framework in the Conclusions section to suggest how to eradicate workplace bullying (lines 311-335).

  1. Resilience is an equivalent of an individual-focused prevention measure that should be considered after addressing the work environment factors (work organization, workload, adequate staffing etc.).

Thank you for your suggestion. We included this in the Conclusions section (lines 328-332).

  1. I am surprised that authors state that few studies have investigated the moderators of workplace bullying. Were synonyms or similar concepts (mobbing, harassment) considered in the research?

To the best of our knowledge, there is still limited research on the moderators of workplace violence including bullying, mobbing and harassment. We included sentences about the similar concepts of workplace bullying, mobbing and harassment in the Introduction section (lines 25-30).

  1. Can authors clarify (row 152) “the nurses had witnessed bullying”: does it include experiencing bullying and witnessing bullying behaviors perpetrated by others to others?

We clarified the sentence as following: the nurses had witnessed bullying behaviors perpetrated by others to others at their workplaces (lines 169-170).

  1. Presentation of data on rows 153 is a bit difficult to read. Is it possible to have the number and the outcome next one to each other?

We revised the sentence as suggested (lines 171-174).

  1. Table 1 would be easier to read with horizontal lines as separators between each variable.

We added horizontal lines as separators between each variable in Table 1 (page 6).

  1. How was the Condition of Work Effectiveness Questionnaire used in the Korean context. Was there a Korean version available? The questionnaire has 12 items, each comprising many sub-items.

The Korean version of the Condition of Work Effectiveness Questionnaire has demonstrated reliability and validity. We provided the reliability and validity information of the instrument in the measurement section (lines 115-117).

  1. Authors state (row 275) that individuals with low resilience are more easily involved and victimized in bullying. Is that data based on longitudinal studies? We can also think that bullied people have a decrease in their resilience level.

The statement was based on the cross-sectional survey study findings. We revised the sentence to tone down the causality (line 296-301).

  1. Where were the questionnaires filled? The location may influence results (at the workplace vs. at home).

The participants completed the questionnaire at any place they wanted. We revised the Participants and procedures section to provide a more detailed description of the data collection procedure (lines 93-98).

  1. Do we have the evidence that providing training on bullying management protocols is effective? Or is that a speculation based on the authors’ work?

There have been research findings on effects of bullying management protocols. We added citations about the bullying management protocols (lines 322-327).

  1. I would like some more description on how the work conditions are assessed, going above the nature of relationships between workers (supplies, medication, beds, patient flows etc.) All of this has a strong influence on psychosocial risks. The Structural Empowerment questionnaire assessed this dimension a little bit, but this seems very limited to me. My understanding is that authors put the emphasis on resilience, training on bullying, bullying monitoring system, education programs (although they mention the need for sufficient staff support).

We measured structural empowerment and perceived workload to assess work conditions. We added both psychological and physical reward strategies which could improve work conditions in the Conclusion section (lines 318-322).

  1. All in all, the questionnaires may be quite long to fill. How long was that? Can authors expand a bit more on missing data (row 86): what is the threshold for them to exclude the 20 cases (e.g. a single non answered question)?

It took 15-20 minutes to complete the questionnaire. We added this information in the Participants and procedures section (line 97).

We expanded description about the missing data as following: We excluded 20 cases with missing data in the study key variables (i.e., experience of workplace bullying, work outcomes, structural empowerment, resilience), and used 435 cases for analysis (lines 100-102).

Reviewer 2 Report

Scope: in the reviewer opinion, the main issue of this manuscript fits into the scope of the IJERPH journal.

Title: it should be more informative – should indicate the type of research approach (cross-sectional correlational study).

Abstract: length and quality are fine.

Keywords: the number of keywords is acceptable.

Introduction: this section is correctly compiled, although it could contain more literature references.

Methods: section 2.2. should provide a more detailed description of the quantitative data collection procedure. The description of the research tools is sufficient. Using a standardized research tools requires providing information about the obtained consent. Did the authors attach the original questionnaire in the supplementary documents to help the readers understand the measurements? What hypotheses did the authors test?

Results: this section was developed in a transparent manner.

Discussion: this section was compiled correctly, although it could be more elaborate (amount of references).

Conclusions: the conclusions are rather postulates (prognostic), than statements.

References: the literature is rather current – a large part of the bibliography items is after 2010 (including 2010). The number of bibliography items is correct. Check the line 399, please.

General comment: the manuscript has been prepared diligently and with attention to detail.

Author Response

Response to Reviewer 2 Comments

Comments and Suggestions for Authors

  1. Scope: in the reviewer opinion, the main issue of this manuscript fits into the scope of the IJERPH journal.

Thank you very much.

  1. Title: it should be more informative – should indicate the type of research approach (cross-sectional correlational study).

Thank you for your suggestion. We amended the title to include the type of research approach as “Moderating effects of structural empowerment and resilience in the relationship between nurses' workplace bullying and work outcomes: A cross-sectional correlational study.”

  1. Abstract: length and quality are fine.

Thank you very much.

  1. Keywords: the number of keywords is acceptable.

Thank you very much.

  1. Introduction: this section is correctly compiled, although it could contain more literature references.

We added more literature references in the Introduction section.

  1. Methods: section 2.2. should provide a more detailed description of the quantitative data collection procedure.

The description of the research tools is sufficient. Using a standardized research tools requires providing information about the obtained consent. Did the authors attach the original questionnaire in the supplementary documents to help the readers understand the measurements?

What hypotheses did the authors test?

We revised the Participants and procedures section to provide a more detailed description of the data collection procedure (lines 93-100).

Before data collection, we obtained approval for the use of the instruments for the research purpose. Unfortunately, we could not provide our questionnaire because the use of the instruments requires the official approvals from the developers and we could not publicize their items without their approvals.

Our study hypotheses were provided (lines 81-84).

  1. Results: this section was developed in a transparent manner.

Thank you very much.

  1. Discussion: this section was compiled correctly, although it could be more elaborate (amount of references).

We added more literature references in the Discussion section.

  1. Conclusions: the conclusions are rather postulates (prognostic), than statements.

To provide our conclusions convincingly, we revised the conclusions section (lines 325-330) and included supporting literature references.

  1. References: the literature is rather current – a large part of the bibliography items is after 2010 (including 2010). The number of bibliography items is correct. Check the line 399, please.

We corrected the typo of the reference number 50 (line 452-453).

  1. General comment: the manuscript has been prepared diligently and with attention to detail.

Thank you very much.